# DNA:RNA Hybrids Are Major Dinoflagellate Minicircle Molecular Types

**DOI:** 10.3390/ijms24119651

**Published:** 2023-06-02

**Authors:** Alvin Chun Man Kwok, Siu Kai Leung, Joseph Tin Yum Wong

**Affiliations:** Division of Life Science, The Hong Kong University of Science and Technology, Clearwater Bay, Kowloon, Hong Kong, China; alvink@ust.hk (A.C.M.K.); leungsiukai82@yahoo.com.hk (S.K.L.)

**Keywords:** DNA:RNA hybrids, minicircle, non-coding region, plastids, chloroplasts

## Abstract

Peridinin-containing dinoflagellate plastomes are predominantly encoded in nuclear genomes, with less than 20 essential chloroplast proteins carried on “minicircles”. Each minicircle generally carries one gene and a short non-coding region (NCR) with a median length of approximately 400–1000 bp. We report here differential nuclease sensitivity and two-dimensional southern blot patterns, suggesting that dsDNA minicircles are in fact the minor forms, with substantial DNA:RNA hybrids (DRHs). Additionally, we observed large molecular weight intermediates, cell-lysate-dependent NCR secondary structures, multiple bidirectional predicted ssDNA structures, and different southern blot patterns when probed with different NCR fragments. In silico analysis suggested the existence of substantial secondary structures with inverted repeats (IR) and palindrome structures within the initial ~650 bp of the NCR sequences, in accordance with conversion event(s) outcomes with PCR. Based on these findings, we propose a new transcription-templating-translation model, which is associated with cross-hopping shift intermediates. Since dinoflagellate chloroplasts are cytosolic and lack nuclear envelope breakdown, the dynamic DRH minicircle transport could have contributed to the spatial-temporal dynamics required for photosystem repair. This represents a paradigm shift from the previous understanding of “minicircle DNAs” to a “working plastome”, which will have significant implications for its molecular functionality and evolution.

## 1. Introduction

Organelle genome transcription-replication-segregation dictates energy transfer within the biosphere and is often coordinated with nuclear genome dynamics. Endosymbiotic organelles, such as chloroplasts, play crucial roles in supplying energy and cellular metabolism, including sugar synthesis, energy storage, and the production of essential amino acids, lipids, and pigments [1,2,3,4]. Peridinin-containing dinoflagellates are the major groups of harmful algal bloom agents and coral symbiotic partners that serve as primary producers in reef ecosystems [5,6,7,8,9]. The peridinin dinoflagellate plastomes present an important case in evolution, as most chloroplast genes have been transferred to the nuclear genome [10,11,12,13,14]. As a result of this transfer, only 15–20 essential plastid genes (compared to over 110 genes on plants’ chloroplast genome [4,15]) remain on “minicircles”, which have been considered as small single-gene circular DNAs of approximately 2 kbp. Among these genes, *psbA* is required for the sustained replacement of the Photosystem II (PSII) component [16,17]. The PSII protein D1, encoded by the *psbA* gene, is continuously damaged during photosynthesis [17], and is regarded as a key component in photoresponses [18,19], with PSII damage identified as an early sign of coral bleaching.

Plastome operations form the basis of primary productivity and underpin all reef ecosystems, in addition to the global abundance of dinoflagellates in aquatic ecosystems. Despite their ecological profoundness, little is known about the molecular biology of minicircles, including the controversy surrounding their localization within the nucleus or chloroplasts [20,21]. Understanding the molecular forms of minicircles is critical to comprehending the dynamics of primary productivity, as dinoflagellate plastid proteins require vesicular transport for targeting [22]. This transport is particularly critical in zooxanthellae, where chloroplasts are pan-plasmamembrane located [23]. The minicircle non-coding regions (NCRs) are less than 1 kbp in length and consist of inverted repeats (IRs) and tandem repeats (TRs). Unlike the plant chloroplast NCRs [24], there is no conservation of replication origin. Previous studies have reported the presence of both long DNAs and RNAs in minicircle preparations [25].

Inspired by newly available reagents, we have continued our studies of the minicircle molecular forms and the implications of the predicted secondary structures. We report here the presence of DNA:RNA hybrid (DRH) forms with minicircle non-coding region (NCR) secondary structures that are predicted to be nicking sites in vivo. Significantly, the commonly regarded dsDNAs are, in fact, a minor minicircle form, representing a paradigm shift in all aspects of minicircle biology.

## 2. Results

### 2.1. Minicircles Contain DNA:RNA Hybrids (DRHs)

The chemical forms of organelle genome nucleic acids, as in the case of minicircles, dictate their mechanistic operations. Inspired by new in silico analysis software and the development of DRH-specific nuclease [26], we investigated the nature of minicircle DNAs (Figure 1). Surprisingly, treatment with RNaseH and RNaseA, which digest the RNA strand in DRHs but not dsDNA, resulted in increased southern blot signals (Figure 1B). This can be explained by the removal of the RNA strand in DRH, which would expose more binding sites (the complementary DNA strand) for the single-stranded southern probe. However, ExoIII digestion, which also digests nicked DNAs, led to increased signals around 2.5–4 kbp. ExoIII (which digests dsDNA and DRHs [27,28]) did not efficiently digest the RNA strand of DRHs or ssDNAs. Large molecular weight (m.w.) intermediates (LMWI > 7 kbp) remained in all nucleases-treated lanes (except Duplex-Specific Nuclease, DSN), despite less efficiency in southern transfer.

Treatment of total nucleic acid preparation with Duplex-Specific Nuclease (DSN, crab nuclease), which degrades dsDNAs and the DNA strand in DRHs [29], completely eliminated all 1.65–4 kbp-southern signals but left a weaker ~500 bp signal streak, likely to be the ssRNAs of the original DRH duplexes. These suggested that the major forms of minicircles are composed of DRHs (double-stranded) and dsDNA (Figure 1B, Lane 4). Since neither RNase A nor RNase H digests ssDNAs or dsDNA, the resulting increase in signals could also be explained if a substantial amount of minicircles were released from inter-circle connections. The lack of streaks was not consistent with having cross-minicircle NCR interactions, although it would be difficult to prove. The ExoIII-treated samples apparently retained more LMWI than other nucleases, and surprisingly more than the untreated sample control. This reconfirmed that substantial LMWI were in DRHs with ExoIII partial digestion of the RNA, although the DRHs could be in multiple formats, including different R-loop types, that could have been retained in the well and likely not accessible to probes. Another possibility is inter-/intra- minicircle or cross-gene-circle interactions (see later discussion). In all possible explanations, the dsDNA circular form, which was previously considered to be the major minicircle form, was found to be a minor fraction.

### 2.2. NCR Mediated Multiple Conversion/Crossing-over Events That Were Restricted after Incubation in the Dinoflagellate Cell Lysate

DRH regions will be resistant to subsequent ExoIII digestion, which could be revealed with southern blot analysis. Simple IRs and palindromic dsDNA repeats will not generate resistance. The 0.4 kbp (L), 0.6 kbp (R), and full 1.1 kbp NCR probes (Figure 4 of [30]) gave similar tracks with different intensities (Figure 4 of [30]). The 0.4 kbp probe produced a stronger southern intensity with larger-size positive electrophoretic mobile types (EMTs) than the 0.6 kbp probe, suggesting that the 3’-region of NCR (678-1102 bp) had a higher representation with larger EMTs (0.4–0.6 kbp) and was likely the 5′ *psbA* gene extension involved in recombination events. The increase in intensity of the 0.3 kbp EMTs (arrowed, and larger-size EMTs) was not in focused spots but spread along the track, confirming this point.

The distinct *psbA* probe only produced a low-intensity spot in the upper track (Figure 4D of [30], ~4 kbp) when compared to the NCR probe, confirming the crossing-over events (when compared with Figure 4A–C of [30]). This observation suggests that the ExoIII-resistant portions were DRHs of NCR without any *psbA* ORF (open reading frame). The large and high-intensity 2 kbp round EMT, which was present in the NCR southern, was indicative of the circular minicircle. In the absence of crossover events, there will be no modular increase with DRH moieties, indicating that the DRHs may have a fostering role.

Binding activity was involved in some transcription-mediated recombination base propagation [31], which could have fostered a distinctive mobility shift or/and multiple PCR products, which would be testable with different template-length PCR. Strikingly, a clean, distinct electrophoretic mobility shift was observed (Figure 2), confirming the presence of a specific factor with high affinity in the *Heterocapsa* (Ht) cell lysate. Non-specific DNA binding protein commonly generates electrophoretic mobility shifts with streaks, whereas specific binding manifests as specific mobility retardation or as a specific gap in PCR. The cell lysate nonetheless contained DNase activity upon further incubation, indicating that the binding factor did not offer protection. The stoichiometry of resistance, however, could have indicated the presence of additional factor(s) that mediated the NCR “opening” that became accessible to the cellular nuclease, as the 1:5 lysate dilution did not exhibit significant degradation (last lane) (Figure 2B). In addition to a potential protein factor, RNA complementary to the NCR regions could have produced a similar gel mobility shift.

This binding-mediated secondary structure stabilization could potentially promote strand conversion events that favor different polymerase readthrough, resulting in unexpected PCR products. In this case, the NCR templated high intensity PCR products with smaller sizes (Figure 2A, before incubation) as well as streaks with very high molecular weight. Incubation with cell lysate reduced the number of smaller products, suggesting that specific binding coerced specific conversion events (Figure 2A, after incubation), and that the secondary structural dynamics were sufficient in driving tandem crossover events, especially considering that the streak extended beyond 6 kbp (even longer when considering the dsDNA size marker) and the inefficiency of Taq polymerase at a longer extension.

### 2.3. Distinct Sized PCR Products with Immobilized PCR Corresponded with Predicted Secondary Structure Locations and Potential Sites for Recombinations

In silico analysis suggested the formation of extensive hairpins that create two tandem palindrome-like repeats at the site between the two 9G repeats (Figure 3E). Both repeats branched essentially at the same site, as predicted by the shortened PCR product with cell lysate incubated PCR (Figure 2A). Nearly all the IR (palindromes) are located within the first ~650 bp of the NCR sequences. This overlaps with the position of clover structures as shown in Figure 3E (bear in mind that Figure 3E is the antisense strand, from 320 to 960 bp).

Despite the vastly different predicted conformations, the predicted NCR ΔG exhibited little changes, and the ΔS (-ve) was essentially three times that of ΔH, indicating that the predicted secondary structures are highly stable but likely shifting between different structures (with little ΔG differences) with 3′ transcription activity. The provision of two putative clover-like substructures that cumulate to one large secondary structure are potential inter- and cross- minicircle interactomes, as well as with ribosomal RNAs. 

In the R+F1 primer PCR with the immobilized template, a non-streak 830 bp product was observed, which was confirmed to be the full NCR plus the 5′-region (first 556 bp) of the *psbA* gene. This suggests that immobilization favored a single secondary structure, as there were no PCR streaks with high resolution (Figure 3A–C). This differed from the ~630 bp PCR product with the free template, suggesting two potential modes of secondary structure-mediated events, with “cross-over” promoted with free ends (i.e., likely non-transcriptional) and transcriptional “restraint”.

The predicted location of eukaryotic-type promotors, situated between the two 9G repeats and exhibiting palindromic-like structures (Figure 2 and Figure 3), implicates the possibility of transcription occurring within the nuclear compartment. The presence of multiple prokaryotic transcription promoters, located with different shifts of the predicted two-clover thermodynamic secondary structures, is consistent with adaptable transcription originating from the other organelle genome. 

### 2.4. Immobilized Template Generated Discrete Double Strand and Single Strand PCR Products

As crossover, but not conversion, requires free ends, it provides an assay to test this hypothesis. We deployed a bead-NCR approach, which would have restricted crossing-over events requiring two free ends. NCR of *psbA* gene was immobilized on gel beads and incubated with *H. triquetra* cell lysate. After washing, the DNA on the beads was subjected to PCR, resulting in the amplification of a 0.6 kbp fragment (Figure 2A). Sequence analysis revealed that a region ~0.4 kbp between the 108 bp direct repeat was absent, indicating that bidirectional dynamics were required for promoting crossing-over events. Furthermore, single-strand primer PCR generated extension with higher sizes (Figure 3B,C), amid weaker in ethidium bromide staining, supporting the provision of ssDNA products with readthrough-type polymerization. This indicated that restriction, likely through transcription-mediated structure, including the palindromic structure (Figure 2B and Figure 3A,E), facilitated readthrough. With numerous tandem repeats, palindrome repeats (when spacer length = 0), and IRs (Figure 3A), NCRs would be expected to form secondary structures. The palindromic structure also gave two single-strand 9G repeats (green arrow) with complementary orientation. These are potential sites for transient binding and resulting in readthrough. IRs and palindromic sites, including those of the CRISPR-Cas9 [32], are recognized sites with high recombinogenic potentials. Since there are predicted promotors in either direction (Figure 3A), transcription-mediated supercoiling ahead of RNA polymerase can potentially generate stresses with strand breakage.

## 3. Discussion

Chloroplasts are highly dynamic organelles that require substantial biogenesis-regeneration responses to photodamage and photoadaptation. Palindromic and inverted repeats serve as hotbeds for secondary structure-mediated molecular events, such as double strand breaks and recombination [33]. Additionally, concomitant transcription and multiplication will foster multiple permutations of DNA:DNA and DNA:RNA interactions. 

The presence of significant DRH and LMWI levels will affect interpretations of previous analyses of minicircle intermediates (Appendix A, Figure for inspection (in non-published material), Figures 4, 5, and 8 of [30]). The *psbA* and NCR probes gave superimposable southern blot tracks in Two-dimensional-gel southern analysis of ExoIII-treated nucleic acid preparations (Figure 5B of [30]). This indicated that each basal spot contained both the NCR and the encoded gene, likely with dsDNAs, suggesting the occurrence of crossing-over events. Despite exhibiting faster mobility, the upper and lower tracks showed similar inter-distances of ~0.5 kbp (Figure 5B of [30]), which did not correspond to the *psbA* dsDNA ORF but rather to the modular increase in poly-ORF DRHs with partial NCR. The superimposition of the southern blots between *psbA*-probed and NCR-probed patterns (Figure 5B of [30], Appendix AA–C) with the NCR probe giving additional larger spots (“B” spots), revealed that the NCR-DRH operatives were dominant, and the *psbA* minicircle was overrepresented, despite over 15 different minicircles being present. The modular ladder (track) shift to the right upon ExoIII treatment was consistent with the removal of nucleic acid extension (nicked, linear extension of dsDNA, Appendix AA), as the two tracks displayed different slopes (Appendix AB,C) converging on destined spots (~2.12 kbp) that were likely dsDNA forms. This suggested that dsDNAs were not a major molecular format. The similar modularity of the upper and lower tracks (~0.4 kbp for dsDNA or ~ 0.8 kbp for ssDNA) indicated that a portion of the NCR was resistant to the nuclease.

The ExoIII trimming of DRHs’ RNA extensions suggests that the initial DRHs did not contain exact complementary lengths of ssRNAs and ssDNAs, which would have formed modular bands on one-dimensional gel. The modularity with different upper-lower slopes suggested crossover events likely occurred during the DRHs’ formation with either of the complementary strands. The faster, unresolved intermediates that did not correspond to Track A or B but were positive in either *psbA* or NCR southern blots (Figure 5B of [30], Appendix A, Figure for inspection-green highlights (in non-published material)) were probably fragments resulting from processing. The lesser efficiency of the LMWI in southern transfer will make comparative analysis difficult, especially considering the presence of the R-loop.

As he size of the 2–3 basal spots (~4 kbp) resemble 2 × minicircle sizes, we also consider the possibility of “hopping + looping” in our model. DRHs, including R-loops and other conformations, do not unequivocally exclude other RNA formats. For example, antisense RNA regulation of *psbA* expression was reported, including dsRNA and associated cyanobacteria, plant, and cyanophage photosynthesis [34,35,36,37,38]. The observation of multiple predicted promotors, likely in opposite orientations, despite the absence of a Shine–Dalgarno (AGGAGGTAAATAATG) sequence, was reminiscent of back-to-back prokaryotic-like promoters within the chloroplast D-loop [39]. Based on our observations, we propose new models (Figure 4) for minicircle multiplications. We aim to explore a variety of possibilities without invoking extensive DSBs, which are likely to occur with bidirectional polymerases. Complementary RNAs will further add to the complexity, including structural and transcriptional roles, which we will address in future investigations. Firstly, the majority of minicircle forms were non-dsDNA DRHs. Secondly, there was a substantial fraction of crossover events. Thirdly, the lower and higher m.w. forms were superimposable, indicating template-based increase. Fourthly, upper-lower tracks were modular, indicating two different modes of multiplication with different nucleic acid forms. Our working model involves transcription-templating coupled with translation. The paradigm shift is that the “minicircles” are multiple nucleic acid plastomes, and the multiple conformations are subjected to prevailing ionotropic photosynthetic conditions and conditioning. Our diagrammatic representations are unlikely to be exhaustive in terms of the combinatory events, particularly considering potential inter- and intra- minicircle interactions.

Previous RT-PCR using primers between the HtNCR and Ht*psbA* gene generated the predicted size product [25] and was consistent with our current finding of DRHs. However, the isolation of RNAs would eliminate secondary ssDNA structures. Considering DRHs, previous interpretations of long polycistronic transcripts [25] are likely associated with the transcript-template complexes, which comprise different combinations as described. The readthrough by polymerase is also different from the generic bacteriophage “rolling-circle” type replication mechanism [30], and our working model is best described as transcription-templating with an inclination towards translation. 

The DRH secondary structures, which has received less attention, is likely play prominent roles in the determination-coupling process through transcription-templating-translation (TTT), which is a better description of transcription-replication-translation. Dinoflagellate genomes contain the plastid 23S rRNA, including minicircle-encoding in some species. *Ht*NCR drove gene expression in the heterotrophic dinoflagellate *Crypthecodinium cohnii* [40], implicating that translation occurs outside of the chloroplast, likely in the mitochondria. High light responsiveness in plant cells was correlated with ribosome occupancy on *psbA* transcripts [15]. DRH would have influenced the efficiency of in situ hybridization studies based on dsDNA plastomes. This could account for the controversy surrounding their subcellular localization(s) [20,21]. Further studies will need to address technical difficulties in resolving DRHs for probe accessibility; the proposed “working plastome” may serve as a chloroplast genome transport agent.

Dinoflagellate plastid proteins were hiked with the vesicular transport system [22]; further studies should address this issue more thoroughly. The *psbA* level, an important facet in photoadaptation and resistance to stress, is light-regulated [41]. The coordination between transcription-conversion-templating and translation in response to light input is an area related to DRH stability that requires further investigation. Chloroplast genome replication could be ROS-regulated independently of organelle segregation [42]; however, it would necessitate a coordination mechanism to ensure proper distribution. In the absence of a membrane potential-dependent MinC-D operative or a FtsZ-mediated division, which are retained in some chloroplasts [43,44], many plastomes will require minimalist solutions for segregation-allocation. The substantial species-level NCR sequence homology suggests the possibility of inter-minicircle conversions, consistent with high m.w. minicircles. This could lead to the formation of “concatenated” minicircles until they are released by inhibitor-sensitive cellular topoisomerases. The association of the *psbA* gene with large m.w. EMTs supports this notion [45]. Understanding whether long DRHs exhibit “coiling” propensity with dsDNA helix parallel potential, which in high supercoil density will exhibit periodicity of semi-open helix, will be crucial to understanding their importance in general molecular events. With photosynthesis being a high-energy environment with intense ionotropic passages, it is conceivable that the differential DNA-RNA charge density will play a critical role in the TTT operatives. The provision of tRNAs, which are encoded in some minicircles, will further enlist the system to mission a supply–demand chain for photosystem repair.

The presence of substantial minicircles in DRHs prompted us to reinterpret our previous results of aphidicolin treatment, which indicated that the unnormalized southern blot differences (previously shown in Figure 3 of [30]) could have been influenced by differences in the genomic DNA level. The effects of thecal plates on diffusion of chemical inhibitors may have also contributed to the negative effects of some inhibitors (e.g., aphidicolin), which should be taken into account when interpreting results. 

Dinoflagellate plastid genomes, despite being highly reduced in size and gene content, offer significant insights into the ecological niche and distribution of these organisms. The interpretation of our results will have broader implications for the evolution of plastids and for key areas of their transcription, multiplication, translation, and segregation, which remain enigmatic even though they are essential for global primary production and carbon fixation.

## 4. Materials and Methods

### 4.1. Culture of Dinoflagellates

*Heterocapsa triquetra* (CCMP 449) was obtained from the Provasoli-Guillard Culture Center for Marine Phytoplankton (CCMP) and maintained in a modified seawater medium (f/2) [46]. The medium has an average salinity of 30‰ and the culture was kept at a constant temperature (18 ± 1 °C) in a 12-hr light/12-hr dark cycle using white fluorescent light. The start of the light period was defined as T = 0, while the start of the dark period was defined as T = 12. The culture was grown to a cell density of ~15,000 cells mL^−1^ as measured by a hemocytometer. 

### 4.2. Nucleic Acid Extraction

All molecular biology manipulations and minicircle reagents essentially followed our published protocol [30], except the deployment of additional nucleases in the analysis.

Genomic and minicircle DNA from *H. triquetra* was prepared using a modified CTAB extraction method [47]. Cells were harvested by low-speed centrifugation (1500× *g*, 10 min), and the cell pellet was resuspended in CTAB extraction buffer (2% [*w*/*v*] CTAB, 2% [*w*/*v*] polyvinylpyrrolidone (PVP-Mr10000), 2 M NaCl, 20 mM EDTA (pH 8.0), 100 mM Tris (pH 8.0), 10 µg mL^−1^ RNase A, 0.1 mg mL^−1^ Proteinase K, 5% [*v*/*v*] β–mercaptoethanol) and incubated at 60–65 °C overnight. The suspension was then extracted with an equal volume of phenol:chloroform:Isoamyl Alcohol (25:24:1) (GE HealthCare, Chicago, IL, USA). After centrifugation (4000× *g*, 20 min), the aqueous phase was transferred to an Eppendorf and mixed with a 2X volume of 100% ethanol. DNA was precipitated by micro-centrifuge at 15,000× *g*, 4 °C for 20 min. 

PCR amplifications were performed to generate minicircle NCR and gene fragments for Southern blot analysis of minicircles and primer extension assays. The primer pairs used in this study are listed in Appendix A. The PCRs were run as follows: 94 °C for 2 min, 30 cycles of 94 °C for 1 min, annealing temperature (stated in Appendix A) for 1.5 min, 72 °C for 2 min, and a final extension step at 72 °C for 10 min.

EC 3.1.27.5: RNase A is an RNase specific for single-stranded RNAs, cleaving at the 3’-end of unpaired C and U residues [5]. EC 3.1.26.4: RNase H is a ribonuclease that cleaves the RNA in a DNA/RNA duplex to produce ssDNA [7]. Duplex-Specific Nuclease (EC 3.1.30.2, DSN, crab nuclease) is an enzyme purified from hepatopancreas of Red King (Kamchatka) crab, which displays a strong preference for cleaving dsDNAs and the DNA strand in DRHs, compared to ssDNAs and ssRNAs [26,29].

### 4.3. Primer Extension Assay

PCR amplifications used *H. triquetra* whole DNA as a template and either Ht-*psbA*-NCR-F or Ht-*psbA*-NCR-R primers under similar conditions as amplifying *psbA* NCR. PCR products were resolved on 6% denaturing polyacrylamide gel with urea [48] at ~20 V cm^−1^ for 2 h. Single-stranded DNA was expected to become denatured in urea and run according to their sizes. DNA in the gel was stained with ethidium bromide and visualized under UV-light. Two-dimensional-gel electrophoresis protocols were identical to our previous report [30].

### 4.4. Gel Retardation Assay and Immobilizated Minicircular DNA as PCR Template

Bead-attached NCR DNA was used as a PCR template to investigate the possible effects of dinoflagellate lysate on the secondary structure of NCR. First, 1.1 kbp *psbA*-NCR was covalently linked between its 5′-phosphate group and the solid base gel-beads (CarboxyLink Coupling Gel, PIERCE) by following the manufacturer’s instructions, generating DNA-bead complexes. CarboxyLink™ Coupling Gel (Pierce Biotechnology, Waltham, MA, USA) is a white gel slurry that contains reactive primary amines to which molecules can be conjugated for use in affinity purification on a gel support. The DNA-bead was then incubated with *H. triquetra* lysate for 2 h at 18 °C, followed by washing with PBS three times. The washed DNA-beads then acted as DNA templates for PCR amplification of *psbA*-NCR using Ht-*psbA*-NCR-F and Ht-*psbA*-NCR-R primers. PCR products were gel-electrophoresed and visualized under UV-light after ethidium bromide staining. The PCR product of interest was cloned into the pGEM-T Easy vector for sequence analysis. DNA fragments of 1.1 kbp *psbA* NCR, 0.6 kbp actin, and 0.7 kbp *psbA* gene were also used to test whether the *H. triquetra* lysate can retard their migration during electrophoresis. Their migration profiles were detected using southern blotting using their corresponding fragments.

### 4.5. In Silico Analysis

Analysis of inverted repeats (IRs) was performed using Palindrome Analyser (version 2.6.6; http://palindromes.ibp.cz/#/en/palindrome; date accessed: 5 April 2023) [49]. The default settings were modified to report inverted repeats with 0 mismatch. Stem sequence lengths were limited to 6–200 bp, and loop (spacer) lengths were limited to 0–1100 bp (mismatch: 0, 14 bp per segment). The IRs and palindrome sequences are listed in Appendix A.

Repetitive DNA sequences were identified using the “Find repeats” function in Unipro UGENE software v.45.1 [50]. The parameters for the “Find repeats” utility were as follows: window size: 11 bp, minimum identity per window: 100%, minimum distance between repeats: 0 bp. The repetitive sequences identified are listed in Appendix A.

Candidate promoter predictions were performed using the Neural Network for Promoter Prediction (NNPP) version 2.2 (https://www.fruitfly.org/seq_tools/promoter.html; minimum promoter score: 0.9, type of organism: eukaryote or prokaryote; date accessed: 5 April 2023), which is based on a neural network of the structural and compositional properties of eukaryotic and prokaryotic core promoter region [51].

The lowest free energy secondary structure of the minicircle *psbA*^NCR^ was generated using the program Quikfold on the DINAMelt sever [52] (http://www.unafold.org/Dinamelt/applications/quickfold.php; DNA at 28 °C, [Na^+^] = 1000 mM, Structures: 5% suboptimal, window size = default; date accessed: 5 April 2023).

## Figures and Tables

**Figure 1 ijms-24-09651-f001:**
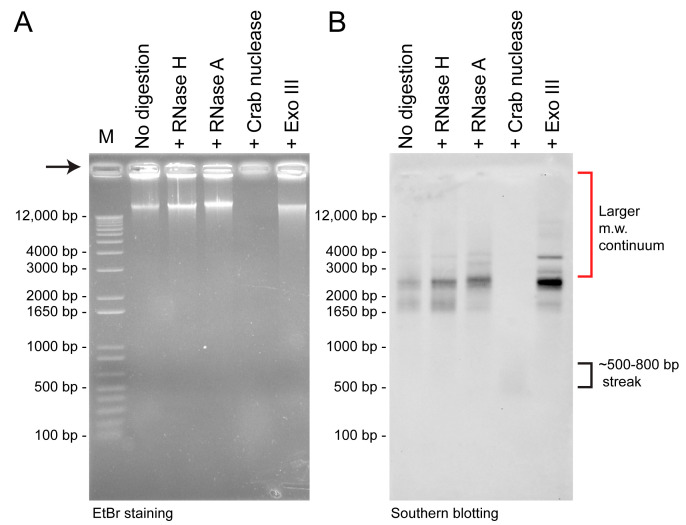
Differential specific digestion of minicircle nucleic acids. (**A**) Ethidium bromide-stained gel of equal loaded total nucleic acid extracts after nuclease digestions. (**B**) Southern blot of (**A**) with *Heterocapsa triquetra psbA*-NCR probe. 10 µg each of total nucleic acid preparations from *H. triquetra* were pre-digested for 1 hr correspondingly with 20 U RNase H (37 °C, Takara), 20 µg mL^−1^ RNase A (37 °C, Roche), 1 U crab DRH/dsDNA-specific nuclease (65 °C, evrogen), or 100 U Exonuclease III (37 °C, NEB). M = DNA marker. Extraction, without pretreatment of RNase through thecommon CTAB method, was performed with the hybridization probe. For 500 bp dsDNA (330 kDa)–800 bp dsDNA (528 kDa), it corresponds to ssRNA of approximately 945–1516 bp in length (with the formula: M.W. of dsDNA = (# nucleotides × 607.4) + 157.9 and m.w. of ssRNA = (# nucleotides × 320.5) + 159.0). It is generally less efficient in transferring large m.w. DNAs over 7- kbp.

**Figure 2 ijms-24-09651-f002:**
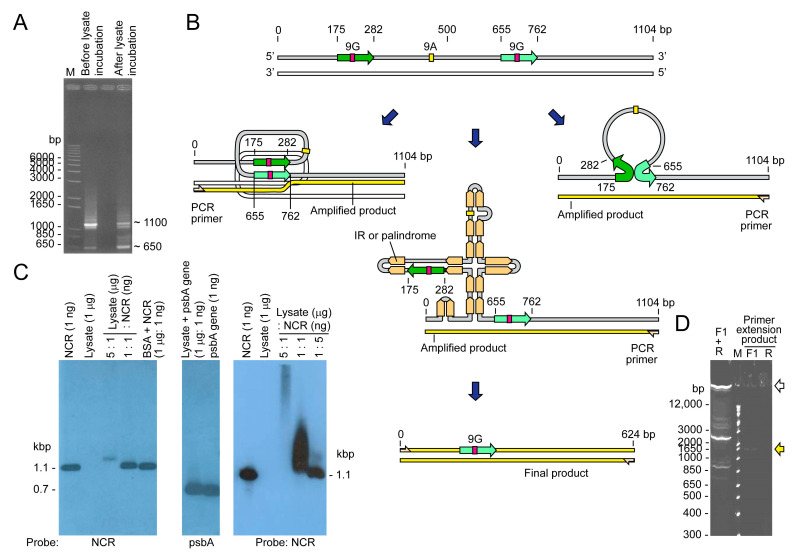
NCR secondary structures manifested with polymerase chain reactions. (**A**) NCR-conjugated–beads were incubated with *H. triquetra* cell lysate before PCR amplification with the primer F1 and R. The ~1.1 kbp band was the full-length NCR amplified product. The ~0.65 kbp band (and larger size streak) became less intense after lysate incubation, and additional bands appeared between 650–1150 bp, indicating some factors in the lysate causing additional structures or crossing-over within the NCR. Sequencing analysis revealed this ~0.65 kbp fragment was in fact a 480-nt truncated *psbA* NCR in which the whole 9A and one 9G conserved cores were absent. A 108 bp repeat sequence (Green arrow) was found to remain in this truncated NCR. Therefore, the secondary structure of NCR fostered as read-through products (including between 650–1150 bp) that were further enriched with incubation in cell lysates. (**B**) Diagrammatic representation of the secondary structure predicted from the PCR results, and through complementation of repeats in opposite directions. The two in-direction 9G repeats could have potentially formed infra-molecular conversion events with single-strand product (as indicated), or as open single-strand regions in palindromic-IR structures that may promote inter- and cross-minicircle interactions. There are also putative complementary RNAs to the NCR that stabilize the shortened region in cells. (**C**) Gel shift assay with *Heterocapsa triquetra* cell lysates. NCRs form secondary structures that were recognized through cellular fraction. The right panel showed treatment with extended incubation time (30 min) indicating potential nuclease activity in the cell lysate. (**D**) Control PCR using template plasmid pGEMT-Ht-*psbA*-NCR, with no cell lysate incubation. The very high m.w. products, which were absent in the immobilized template, would have involved inter-minicircle interactions. The distinctly weaker products with F1 or R single primer PCR were consistent with single strand extension; the high m.w. product suggested readthrough to rolling-circle type product. The distinct difference in PCR products suggested immobilization on one side leading to different PCR products.

**Figure 3 ijms-24-09651-f003:**
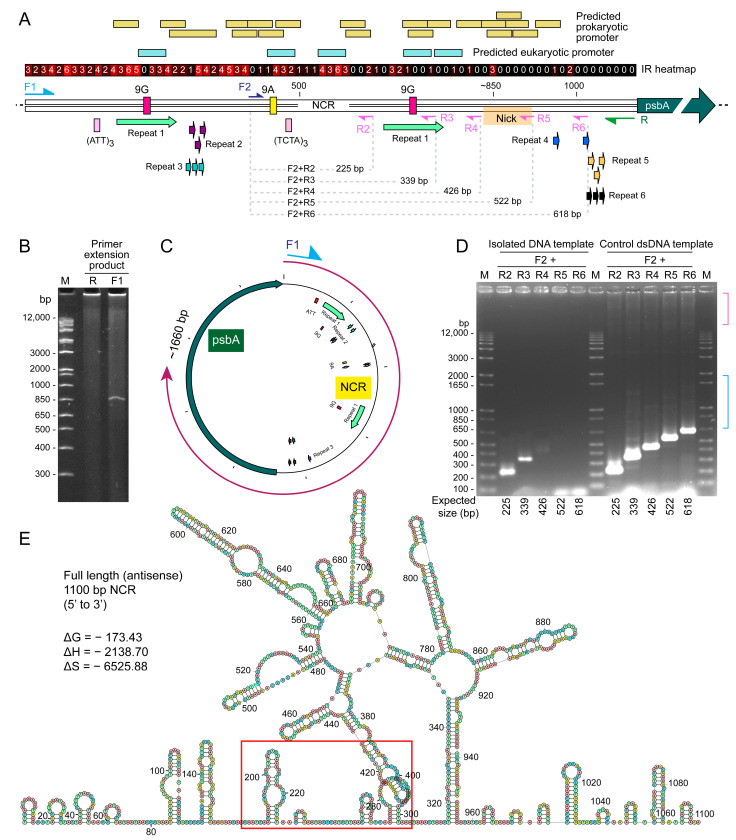
Single-stranded PCR suggested a polymerase run-through at predicted secondary structure. (**A**) A schematic diagram showing the relative position of PCR primers used in PCR extension assay and the following nested PCR, for verifying the location of the possible nick site on NCR. Location of the predicted prokaryotic and eukaryotic promoters, and inverted repeats (IR) were shown. The IR heatmap suggested that most of the IRs were located at 5′-end of NCR (3′-end of NCR). Details of palindrome and IR positions are listed in Appendix A. Details of repeat sequences are listed in Appendix A. (**B**) Primer extension assay of *psbA*-minicircle NCR. PAGE analysis of single-stranded extension products amplified using either single oligo-primer F1 and R. A band at ~850 bp was observed when using primer F1, but not with primer R. (**C**) The intensity of discrete F1 product appeared to be double strand around 830 base pairs, indicating the tandem palindromic repeats reverse the PCR direction. Noted the diffuse products exhibited extended size distribution to the origin, indicating potential branching products. (**D**) No nested PCR products were generated using the reverse primers R5 and R6, with the isolated template (left panel) suggesting that the palindromic site exerted reverse direction at the ~830 bp mark. Using different F2+ combinations with R2-R6 gave expected dsPCR products (right panel) amid with additional, less discrete size products (expected 22 and 339 bp), in agreement with shorter ssDNAs reannealing more promiscuously with additional secondary structures. M = DNA marker. Different upper bands (red bracket) and vertical streak (blue bracket), suggesting discrete reverse direction events requiring different elements. This is surprising as the annealing temperature of the PCR reactions was set at 54 °C (with predicted 42 °C), suggesting that the secondary structure persisted even at this high temperature. Such a secondary structure will operatively promote crossing-over events under non-PCR conditions. (**E**) DNA secondary structure predictions, based on minimum free energy (by Quikfold), of the antisense strand of the NCR. Red box indicates the possible nicking site (in between 180–299 bp on the antisense strand, and the complementary location (806–925 bp) on the sense strand in (**E**)) on the NCR. Only the predicted structures with the lowest free energy ΔG values (kcal mol^−1^) are shown. Nearly all of the IR (palindrome) are located within the first ~650 bp of the NCR sequences. This overlaps with the position of clover structures as shown.

**Figure 4 ijms-24-09651-f004:**
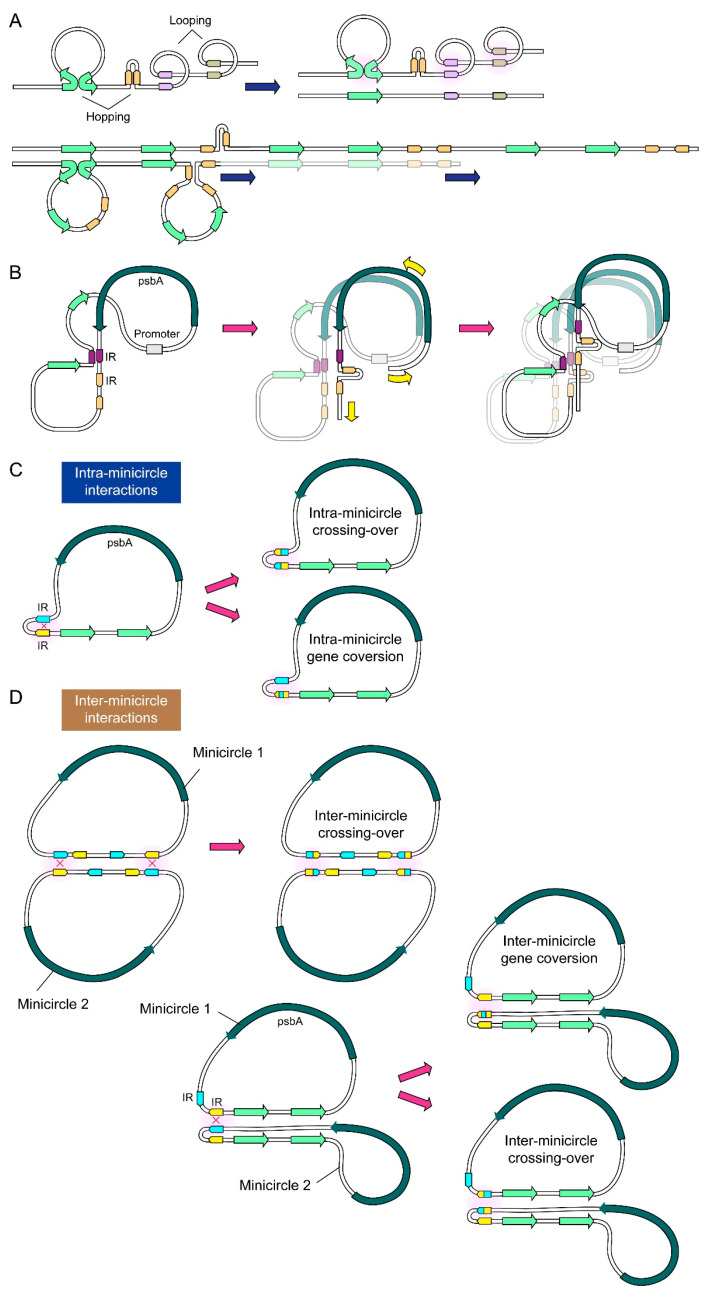
Diagrammatic representation of loop-hopping model for minicircle transcription-multiplication. (**A**) Looping is facilitated by a combination of IRs and TRs, while hopping is enabled by transcription-secondary structures. Only selective topological permutations are represented. (**B**) With DRHs, “loop-hopping” can progress to transcription-templating with conversion events involving palindromic secondary structure assembly (see Figure 3). Possible (**C**) intra-minicircle or (**D**) inter-minicircle interactions are also shown.

## Data Availability

Not applicable.

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
