# Peer review of "DNA:RNA Hybrids Are Major Dinoflagellate Minicircle Molecular Types"

_ijms, 2023, doi:10.3390/ijms24119651_

Round 1

Reviewer 1 Report

Review report of the MS_ijms-2432530-peer-review-v1

The manuscript ‘DNA:RNA hybrids are major Dinoflagellate minicircle molecular type’ Authors by Brychkova et al. is well-written. However, I have a few suggestions which could improve the manuscript.

The manuscript is submitted without line number, hence, difficult to suggest changes.

Abstract: ‘Transcripogenome’ …please check. Pease throw some more light..

Introduction: Instead of mentioning ‘we report’ please mention the clear-cut aim/objective of the work along with the future thrust aligned with the research gap.

Materials and methods:

Please mention the reference ‘Puchooa, 2004’ in numbered format as per the journal standard.

Section 4.2: ‘Our published protocol’.. please cite the reference.

Please revise ‘Whole DNA’ as ‘gDNA’

What is GE Health???

Please write ‘Sambrook and Russell, 2001’ in numbered format.

CarboLink Coupling Gel, PIERCE….Please mention the make, city, and Country of the reagents and equipment used in the study.

‘Unipro UGENE software’..Please mention the date of access for all the softwares and online tools used in the study.

Results: Please focus on the significant finding of the study instead of more discussion. The results should be discussed in the discussion section in stead of result section.

(Figure 4 of [19])…Whether necessary permission has been accorded from [19]?

Discussion: What do you mean by Figures 4, 5, 8 of [19])?

The discussion is satisfactory.

References may be arranged as per the journal pattern. Please cross-check the references cited with the list and text.

Although the overall grammar score is satisfactory, the authors may once polish the language, grammar, and punctuation while doing the minor revision.

The manuscript may be accepted with minor corrections.

Good luck with the revision.

Author Response

We would like to thank all the reviewers for their constructive suggestions and comments that have helped us improve the manuscript. 

We have revised Table S3, Figure 4 (and legend), references, and included additional experimental details as per the suggestions by reviewers. 

Point-to-point replies to the reviewers’ comments are listed as follows: 

Abstract: ‘Transcripogenome’ …please check. Pease throw some more light..

 Changed to “plastome”.

Introduction: Instead of mentioning ‘we report’ please mention the clear-cut aim/objective of the work along with the future thrust aligned with the research gap.

Thanks for the suggestion. To improve, we have added the following statement in the Introduction:

Plastome operations form the basis of primary productivity and underpin all reef ecosystems, in addition to the global abundance of dinoflagellates in aquatic ecosystems. Despite their ecological profoundness, little is known about the molecular biology of minicircles, including the controversy surrounding their localization within the nucleus or chloroplasts [20, 21]. Understanding the molecular forms of minicircles is critical to comprehending the dynamics of primary productivity, as dinoflagellate plastid proteins require vesicular transport for targeting [22].” 

Materials and methods:

Please mention the reference ‘Puchooa, 2004’ in numbered format as per the journal standard.

Amended.

Section 4.2: ‘Our published protocol’.. please cite the reference.

Reference inserted.

Please revise ‘Whole DNA’ as ‘gDNA’

We changed it to “Genomic and minicirlce DNA”.

What is GE Health???

Amended. It’s “GE HealthCare, USA”.

Please write ‘Sambrook and Russell, 2001’ in numbered format.

 Amended.

CarboLink Coupling Gel, PIERCE….Please mention the make, city, and Country of the reagents and equipment used in the study.

To make it more clear, we have added the following statement in 4.4.

“CarboxyLink™ Coupling Gel (Pierce Biotechnology, USA) is a white gel slurry that contains reactive primary amines to which molecules can be conjugated for use in affinity purification on a gel support.”

 ‘Unipro UGENE software’..Please mention the date of access for all the softwares and online tools used in the study.

Date of access have been added for all the online tools used in this study.

Because Unipro UGENE is not an online tools, and so we only include the version number.

“Repetitive DNA sequences were identified using the “Find repeats” function in Unipro UGENE software v.45.1”

Results: Please focus on the significant finding of the study instead of more discussion. The results should be discussed in the discussion section in stead of result section.

 (Figure 4 of [19])…Whether necessary permission has been accorded from [19]?

There is no copyrights issue. Thank you for the reminder.

Discussion: What do you mean by Figures 4, 5, 8 of [19])?

It means the Figures 4, 5 and 8 in the reference 19.

The discussion is satisfactory.

References may be arranged as per the journal pattern. Please cross-check the references cited with the list and text.

 Rechecked.

Although the overall grammar score is satisfactory, the authors may once polish the language, grammar, and punctuation while doing the minor revision.

The manuscript may be accepted with minor corrections.

We believe that the revisions made in response to the reviewer comments have significantly improved the quality and clarity of our manuscript. We are grateful for the reviewer constructive feedback, which has guided us in enhancing the scientific integrity and readability of our work.

Reviewer 2 Report

In this study, the authors provide new information about minicircle DNA and propose a transport mechanism and potential molecular functionality and evolution pattern. Further evaluation of the full paper's arguments and data reliability and scientific value is needed. I encountered aspects in the manuscript that diminish its scientific value and require improvement. Here I provide several comments that may be of utility to the authors.

1. To provide readers with a comprehensive understanding of the importance of chloroplast genomes and their role in cellular function, you can enrich the background description of the chloroplast genome in your paper by referencing the following articles:DOI: 10.1371/journal.pone.0192966, DOI: 10.3389/fpls.2022.832884.

2. Please aim for a more concise and focused Discussion section that highlights the key implications of your study. This will help to improve the clarity and impact of your message.

1. There are several language issues present throughout the paper that may diminish its scientific value. These include incorrect usage of articles (such as "a" and "the"), inconsistent use of singular and plural nouns, and incorrect verb conjugation in some instances.

2. "The 0.4 kbp (L), 0.6 kbp (R) and full 1.1 kbp NCR probes " should be "The 0.4 kbp (L), 0.6 kbp (R), and full 1.1 kbp NCR probes ".

3. The sentence "Fourthly, upper-lower tracks were modular, indicating potential different modes of multiplicaton with different...", "multiplicaton" should be "multiplication".

4. Additionally, I noticed some inconsistencies in the formatting and writing style of gene names such as psbA.

I highly recommend that you carefully check the entire paper for these language issues and make the necessary corrections to ensure accuracy and clarity. Proper usage of grammar is crucial for maintaining scientific integrity and enhancing the readability of the paper.

Author Response

We would like to thank all the reviewers for their constructive suggestions and comments that have helped us improve the manuscript. 

We have revised Table S3, Figure 4 (and legend), references, and included additional experimental details as per the suggestions by reviewers. 

Point-to-point replies to the reviewers’ comments are listed as follows: 

  1. To provide readers with a comprehensive understanding of the importance of chloroplast genomes and their role in cellular function, you can enrich the background description of the chloroplast genome in your paper by referencing the following articles: DOI: 10.1371/journal.pone.0192966, DOI: 10.3389/fpls.2022.832884.

References added.

The following paragraph (underlined) was added in the Introduction:

“………….. Endosymbiotic organelles, such as chloroplasts, play crucial roles in supplying energy and cellular metabolism, including sugar synthesis, energy storage, and the production of essential amino acids, lipids and pigments [1-4]. …………………… As a result of this transfer, only 15-20 essential plastid genes (comparing to over 110 genes on plants chloroplast genome [4, 15]) remain on "minicircles," which are small, single-gene circular DNAs of approximately 2 kbp. …………”

  1. Please aim for a more concise and focused Discussion section that highlights the key implications of your study. This will help to improve the clarity and impact of your message.

We agree with your suggestion to aim for a more concise and focused Discussion section. We have revised the Discussion section, emphasizing the key implications of our study and eliminating unnecessary details. Please check the revised manuscript.

Comments on the Quality of English Language

  1. There are several language issues present throughout the paper that may diminish its scientific value. These include incorrect usage of articles (such as "a" and "the"), inconsistent use of singular and plural nouns, and incorrect verb conjugation in some instances.

We have thoroughly revised the entire paper, addressing incorrect usage of articles, inconsistencies in the use of singular and plural nouns, and incorrect verb conjugation. Please check the revised manuscript.

  1. "The 0.4 kbp (L), 0.6 kbp (R) and full 1.1 kbp NCR probes " should be "The 0.4 kbp (L), 0.6 kbp (R), and full 1.1 kbp NCR probes ".

We have amended the sentence as suggested.

  1. The sentence "Fourthly, upper-lower tracks were modular, indicating potential different modes of multiplicaton with different...", "multiplicaton" should be "multiplication".

Amended.

  1. Additionally, I noticed some inconsistencies in the formatting and writing style of gene names such as psbA.

We have carefully checked the entire paper for inconsistencies in the formatting and writing style of gene names, such as psbA. We have now standardized the gene names throughout the manuscript to ensure consistency.

I highly recommend that you carefully check the entire paper for these language issues and make the necessary corrections to ensure accuracy and clarity. Proper usage of grammar is crucial for maintaining scientific integrity and enhancing the readability of the paper.

We believe that the revisions made in response to your comments have significantly improved the quality and clarity of our manuscript. We are grateful for your constructive feedback, which has guided us in enhancing the scientific integrity and readability of our work.

We hope that the revised manuscript would be acceptable to be published in International Journal of Molecular Sciences.

Reviewer 3 Report

Dear Authors,

It was a pleasure to read your manuscript. "DNA:RNA hybrids are major Dinoflagellate minicircle molecular types".

I feel the manuscript should undergo some minor revision based on the suggestions of the Tables and Figures in a clear presentation.

The manuscript is clear and well written with the support of previous literature on minicircle DNA and also presented in a well-structured manner. All the citations are relevant and support to the study. The manuscript is scientifically sound and contributes to the plastid genome editing field as a minicircle plasmid vector for chloroplast targeting engineering. The work was designed well and executed with a scientific approach, all the results presented in reproducible, and the details given in the methods sections are appropriate. English writing is understandable.

There are some areas for minor revision/suggestions to be considered before consideration for acceptance.

Query 1:

Table S1 Primers and Subsection 4.3. Primer extension assay

 There is lack of PCR-related information, like conditions and cycles and TM’s. Add in detail in the appropriate sections.

Query 2:

Table S3: Repetitive sequences identified on the HtNCR sequences.

The authors only mentioned the number of Repeat sequences from HtNCR.

Revise the table, Column 1 Repeat to Repeats ID R1, R2.... Etc..

The column 5 title cab is being location in Ht NCR

Query 3:

Figure 4. Diagrammatic representation of the loop-hoping model for minicircle transcription-multiplication.

Figure 4, can be split into two Fig. 4A for intra-minicircle interactions and Fig 4B for inter-minicircle interactions. In the present view, there is a lack of self-explanatory.

Author Response

We would like to thank all the reviewers for their constructive suggestions and comments that have helped us improve the manuscript. 

We have revised Table S3, Figure 4 (and legend), references, and included additional experimental details as per the suggestions by reviewers. 

Point-to-point replies to the reviewers’ comments are listed as follows: 

Query 1:

Table S1 Primers and Subsection 4.3. Primer extension assay

There is lack of PCR-related information, like conditions and cycles and TM’s. Add in detail in the appropriate sections.

A new paragraph about the PCR conditions and cycles was added in 4.2. Nucleic Acid Extraction:

“PCR amplifications were performed to generate minicircle NCR and gene fragments for Southern blot analysis of minicircles, and primer extension assays. The primer pairs used in this study were listed in Table S1. The PCRs were run as follows: 94°C for 2 min, 30 cycles of 94°C for 1 min, annealing temperature (stated in Table S3) for 1.5 min, 72°C for 2 min, and a final extension step at 72°C for 10 min.”

And the annealing temperature used in different PCRs were provided in the updated Table S1.

Query 2:

Table S3: Repetitive sequences identified on the HtNCR sequences.

The authors only mentioned the number of Repeat sequences from HtNCR.

Revise the table, Column 1 Repeat to Repeats ID R1, R2.... Etc..

The column 5 title cab is being location in Ht NCR

Table S3 was revised as suggested.

Query 3:

Figure 4. Diagrammatic representation of the loop-hoping model for minicircle transcription-multiplication.

Figure 4, can be split into two Fig. 4A for intra-minicircle interactions and Fig 4B for inter-minicircle interactions. In the present view, there is a lack of self-explanatory.

We have split Figure 4 into Figure 4A-D, with revised Figure captions.

“Figure 4. Diagrammatic representation of loop-hoping model for minicircle transcription- multiplication. (A) Looping enabled by a combination of IRs and TRs. Hoping enabled by transcription-secondary structures. Only selective topological permutations are represented. (B) With DRHs, “Loop-hopping” can progress to transcription-templating with conversion events with palindromic secondary structure assembly (see Figure 3). Possible (C) intra-minicircle or (D) inter-minicircle interactions. This model also explained the previous 2-D gel analysis results of modular spots on tracks, each intermediate can potentially contain DRHs as well as dsDNAs.”

We hope that the revised manuscript would be acceptable to be published in International Journal of Molecular Sciences.

Reviewer 4 Report

Review for the paper “DNA:RNA hybrids are major Dinoflagellate minicircle molecular types”

Dinoflagellates are an abundant and diverse group of chromalveolate algae that, among other ecological roles, are the principal photosynthetic symbionts of corals and contain an extraordinary diversity of chloroplast lineages acquired through a variety of serial secondary and tertiary endosymbioses. Peridinin-containing dinoflagellates contain the most unusual chloroplast genome reported to date, with secondary chloroplasts derived from red algae. The chloroplast genome of the peridinin-containing dinoflagellates is unique in that it consists of a series of small circular DNA molecules known as minicircles. Typically, each minicircle contains a single gene, with a core region that is similar among minicircles within species, but species or strain-specific. There are also minicircles that contain more than one coding region, and minicircles that have a core region, but contain only fragments of the coding sequences, or no coding sequences at all. Minicircles lack recognizable promoter sequences for the nuclear- or chloroplast-encoded polymerases used by plants. However, conserved sequence elements immediately upstream of the translation start sites and in the minicircle cores have been proposed to act as non-canonical promoters. Using novel approaches, the authors investigated the nature of minicircle DNAs in dinoflagellates. The authors' results suggest that dsDNA minicircles are minor forms with substantial DNA:RNA hybrids. They found substantial secondary structures with inverted repeats and palindromic structures and proposed a new working model related to cross-hopping shift intermediates. These data improve our knowledge of the molecular functionality and evolution of the chloroplast genome of peridinin-containing dinoflagellates.

All these reasons explain the relevance of the paper by Alvin Chun Man Kwok and co-authors submitted to "International Journal of Molecular Sciences".

General scores.

The data presented by the authors are original and significant. The study is correctly designed and the authors used appropriate sampling methods. In general, statistical analyses are performed with good technical standards. The authors conducted careful work that may attract the attention of a wide range of specialists focused on dinoflagellate molecular biology.

Recommendations.

P 1. Please, provide citations for this sentence: Peridinin-containing dinoflagellates are the major groups of harmful algal bloom agents and coral symbiotic partners that serve as primary producers in reef ecosystems.

P. 11. Please, provide a salinity level for the seawater used in this study.

P 11. Please, provide citations for this sentence: All the molecular biology manipulations and minicircle reagents essentially followed our published protocol except the deployment of additional nucleases in the analysis.

The authors should discuss the applicability of their results to the entire group of dinoflagellates, as dinoflagellates are common to abundant in both marine and freshwater environments and at both high and low latitudes.

References. The Latin names should be written in italics.

Specific remarks.

p 1. Consider replacing “dictate energy transfer within the biosphere and are often coordinated” with “dictates energy transfer within the biosphere and is often coordinated”

p 1. Consider replacing “as the key component in photoresponses [8, 9], with PSII damages being identified” with “as a key component in photoresponses [8, 9], with PSII damage identified”

p 2. Consider replacing “that are predicted to be nicking site in vivo.” with “that are predicted to be nicking sites in vivo.”

p 2. Consider replacing “Surprisingly, treatment with RNaseH and RNaseA, which digests the RNA strand in DRHs but not dsDNA, resulted in increased the southern blot signals (Figure 1B).” with “Surprisingly, treatment with RNaseH and RNaseA, which digest the RNA strand in DRHs but not dsDNA, resulted in increased southern blot signals (Figure 1B).”

p 2. Consider replacing “could also be explained if substantial amount of minicircles being released from” with “could also be explained if a substantial amount of minicircles were released from”

p 3. Consider replacing “2.2.   NCR mediated mutliple” with “2.2.   NCR mediated multiple”

p 3. Consider replacing “Sequencing analysis revealed this ~0.65 kbp fragment was in fact a 480-nt truncated psbA NCR which” with “Sequencing analysis revealed this ~0.65 kbp fragment was in fact a 480-nt truncated psbA NCR in which”

p 3. Consider replacing “as read-through products (including between 650-1150bp) that was further” with “as read-through products (including between 650-1150bp) that were further”

p 5. Consider replacing “based structural predictions are listed in Figure 3” with “based structural predictions are shown in Figure 3”

p 5. Consider replacing “2.3. Distinct sized PCR products with immobilized PCR corresponded with predicted secondary structure locations, and potential sitse for recombinations” with “2.3. Distinct sized PCR products with immobilized PCR corresponded with predicted secondary structure locations and potential sites for recombinations”

p 7. Consider replacing “This is surprising as the annealing temperature of the PCR reactions were set at” with “This is surprising as the annealing temperature of the PCR reactions was set at”

p 7. Consider replacing “Nearly all of the IR (palindrome) are located within first ~650 bp of the NCR sequences. This overlap with the position of clover structures as shown.” with “Nearly all of the IR (palindrome) are located within the first ~650 bp of the NCR sequences. This overlap with the position of clover structures is shown.”

p 7. Consider replacing “The presence of a signifi cant DRH and LMWI levels will” with “The presence of significant DRH and LMWI levels will”

p 8. Consider replacing “Antisense RNA regulation of psbA expression were reported, including dsRNA and associated cyanobacteria, plant and cyanophage photosynthesis [23-27].” with “Antisense RNA regulation of psbA expression was reported, including dsRNA and associated cyanobacteria, plant and cyanophage photosynthesis [23-27].”

p 10. Consider replacing “will possibly serve as transport agent.” with “will possibly serve as a transport agent.”

p 11. Consider replacing “Culture was grown to a cell density of ~15,000 cells ml-1 as measured by hemocytometer.” with “The culture was grown to a cell density of ~15,000 cells ml-1 as measured by a hemocytometer.”

p 11. Consider replacing “PCR amplifications using H. triquetra whole DNA as template and either Ht-psbA- NCR-F or Ht-psbA-NCR-R primers were run as similar” with “PCR amplifications using H. triquetra whole DNA as a template and either Ht-psbA- NCR-F or Ht-psbA-NCR-R primers were run under similar”

p 11. Consider replacing “Bead-attached NCR DNA was used as PCR template to probe the” with “Bead-attached NCR DNA was used as a PCR template to probe the”

p 11. Consider replacing “followed with PBS washing for 3 times.” with “followed by PBS washing 3 times.”

p 12. Consider replacing “Repetitive DNA sequences were identified using the the "Find repeats"” with “Repetitive DNA sequences were identified using the "Find repeats"”

p 12. Consider replacing “were generated using the program Quikfold on the DINAMelt sever [39]” with “was generated using the program Quikfold on the DINAMelt sever [39]”

Some minor revisions are recommended. 

Author Response

We would like to thank all the reviewers for their constructive suggestions and comments that have helped us improve the manuscript. 

We have revised Table S3, Figure 4 (and legend), references, and included additional experimental details as per the suggestions by reviewers. 

Point-to-point replies to the reviewers’ comments are listed as follows: 

P 1. Please, provide citations for this sentence: Peridinin-containing dinoflagellates are the major groups of harmful algal bloom agents and coral symbiotic partners that serve as primary producers in reef ecosystems.

 Added.

  1. 11. Please, provide a salinity level for the seawater used in this study.

The salinity of the seawater-based medium (f/2) used in this study was 30 ‰.

And we have added this info in the Materials and Methods session:

“Heterocapsa triquetra (CCMP 449) was obtained from Provasoli-Guillard Culture Center for Marine Phytoplankton (CCMP) and maintained in a modified seawater medium (f/2) [35], which has an average salinity of 30 ‰, at constant temperature (18ËšC) in a 12-hr light/12-hr dark cycle using white fluorescent light.”

P 11. Please, provide citations for this sentence: All the molecular biology manipulations and minicircle reagents essentially followed our published protocol except the deployment of additional nucleases in the analysis.

Added.

The authors should discuss the applicability of their results to the entire group of dinoflagellates, as dinoflagellates are common to abundant in both marine and freshwater environments and at both high and low latitudes.

The following paragraph was added at the end of Discussion:

“Dinoflagellate plastid genomes, despite being highly reduced in size and gene content, offer significant insights into the ecological niche and distribution of these organisms. The interpretation of our results will have broader implications for the evolution of plastids and for key areas of their transcription, multiplication, translation, and segregation, which remain enigmatic even though they are essential for global primary production and carbon fixation.”

References. The Latin names should be written in italics.

The Latin names in the references have been rechecked and written in italics.

Specific remarks.

p 1. Consider replacing “dictate energy transfer within the biosphere and are often coordinated” with “dictates energy transfer within the biosphere and is often coordinated”

Replaced.

p 1. Consider replacing “as the key component in photoresponses [8, 9], with PSII damages being identified” with “as a key component in photoresponses [8, 9], with PSII damage identified”

Replaced.

p 2. Consider replacing “that are predicted to be nicking site in vivo.” with “that are predicted to be nicking sites in vivo.”

Replaced.

p 2. Consider replacing “Surprisingly, treatment with RNaseH and RNaseA, which digests the RNA strand in DRHs but not dsDNA, resulted in increased the southern blot signals (Figure 1B).” with “Surprisingly, treatment with RNaseH and RNaseA, which digest the RNA strand in DRHs but not dsDNA, resulted in increased southern blot signals (Figure 1B).”

Replaced.

p 2. Consider replacing “could also be explained if substantial amount of minicircles being released from” with “could also be explained if a substantial amount of minicircles were released from”

Replaced.

p 3. Consider replacing “2.2.   NCR mediated mutliple” with “2.2.   NCR mediated multiple”

Replaced.

p 3. Consider replacing “Sequencing analysis revealed this ~0.65 kbp fragment was in fact a 480-nt truncated psbA NCR which” with “Sequencing analysis revealed this ~0.65 kbp fragment was in fact a 480-nt truncated psbA NCR in which”

Replaced.

p 3. Consider replacing “as read-through products (including between 650-1150bp) that was further” with “as read-through products (including between 650-1150bp) that were further”

Replaced.

p 5. Consider replacing “based structural predictions are listed in Figure 3” with “based structural predictions are shown in Figure 3”

Replaced.

p 5. Consider replacing “2.3. Distinct sized PCR products with immobilized PCR corresponded with predicted secondary structure locations, and potential sitse for recombinations” with “2.3. Distinct sized PCR products with immobilized PCR corresponded with predicted secondary structure locations and potential sites for recombinations”

Replaced.

p 7. Consider replacing “This is surprising as the annealing temperature of the PCR reactions were set at” with “This is surprising as the annealing temperature of the PCR reactions was set at”

Replaced.

p 7. Consider replacing “Nearly all of the IR (palindrome) are located within first ~650 bp of the NCR sequences. This overlap with the position of clover structures as shown.” with “Nearly all of the IR (palindrome) are located within the first ~650 bp of the NCR sequences. This overlap with the position of clover structures is shown.”

Replaced.

p 7. Consider replacing “The presence of a signifi cant DRH and LMWI levels will” with “The presence of significant DRH and LMWI levels will”

Replaced.

p 8. Consider replacing “Antisense RNA regulation of psbA expression were reported, including dsRNA and associated cyanobacteria, plant and cyanophage photosynthesis [23-27].” with “Antisense RNA regulation of psbA expression was reported, including dsRNA and associated cyanobacteria, plant and cyanophage photosynthesis [23-27].”

Replaced.

p 10. Consider replacing “will possibly serve as transport agent.” with “will possibly serve as a transport agent.”

Replaced.

p 11. Consider replacing “Culture was grown to a cell density of ~15,000 cells ml-1 as measured by hemocytometer.” with “The culture was grown to a cell density of ~15,000 cells ml-1 as measured by a hemocytometer.”

Replaced.

p 11. Consider replacing “PCR amplifications using H. triquetra whole DNA as template and either Ht-psbA- NCR-F or Ht-psbA-NCR-R primers were run as similar” with “PCR amplifications using H. triquetra whole DNA as a template and either Ht-psbA- NCR-F or Ht-psbA-NCR-R primers were run under similar”

Replaced.

p 11. Consider replacing “Bead-attached NCR DNA was used as PCR template to probe the” with “Bead-attached NCR DNA was used as a PCR template to probe the”

Replaced.

p 11. Consider replacing “followed with PBS washing for 3 times.” with “followed by PBS washing 3 times.”

Replaced.

p 12. Consider replacing “Repetitive DNA sequences were identified using the the "Find repeats"” with “Repetitive DNA sequences were identified using the "Find repeats"”

Replaced.

p 12. Consider replacing “were generated using the program Quikfold on the DINAMelt sever [39]” with “was generated using the program Quikfold on the DINAMelt sever [39]”

Replaced.

We hope that the revised manuscript would be acceptable to be published in International Journal of Molecular Sciences.